# EEG Alpha Band Responses Reveal Amplification Benefits in Infants with Hearing Loss

**DOI:** 10.3390/children10030600

**Published:** 2023-03-21

**Authors:** Kristin Uhler, Daniel J. Tollin, Phillip M. Gilley

**Affiliations:** 1Department of Physical Medicine and Rehabilitation, University of Colorado Anschutz School of Medicine & Children’s Hospital Colorado, Aurora, CO 80045, USA; 2Department of Physiology and Biophysics, University of Colorado Anschutz School of Medicine, Aurora, CO 80045, USA; 3Institute of Cognitive Science, University of Colorado Boulder, Boulder, CO 80309, USA

**Keywords:** infant speech perception, EEG, event-related potential, mismatch response, alpha

## Abstract

Our objective was to examine the effects of hearing aid amplification on auditory detection and discrimination in **i**nfants who were **h**ard of **h**earing (IHH) using a physiological measure of auditory perception. We recorded EEG from 41 sleeping IHH aged 1.04 to 5.62 months while presenting auditory stimuli in a mismatch response paradigm. Responses were recorded during two listening conditions for each participant: aided and unaided. Temporal envelopes of the mismatch response in the EEG alpha band (6–12 Hz) were extracted from the latent, time-frequency transformed data. Aided alpha band responses were greater than unaided responses for the deviant trials but were not different for the standard trials. Responses to the deviant trials were greater than responses to the standard trials for the aided conditions but were not different for the unaided conditions. These results suggest that the alpha band mismatch can be used to examine both detection and discrimination of speech and non-speech sounds in IHH. With further study, the alpha band mismatch could expand and refine our abilities to validate hearing aid fittings at younger ages than current clinical protocols allow.

## 1. Introduction

In the United States, **i**nfants who are **h**ard of **h**earing (IHH) are fit with hearing aids (HA), on average, between 4–7 months of age [1,2]. The current clinical guidelines recommend the use of real ear probe microphone measures, which quantify the audibility of a given input signal within the context of infants’ hearing thresholds. This quantification is known as the aided speech intelligibility index (aided SII ANSI 1997). During rapid periods of language learning, clinicians rely on this aided SII, which does not directly measure speech discrimination abilities. However, hearing aid validation, whether an infant can differentiate speech sounds, is difficult to assess because of their inability to complete behavioral tasks. Among older children [3] and adults after verification through probe microphone measures [4], validation can be completed using behavioral speech discrimination measures. In addition to behavioral measures, several studies have used passive electroencephalography (EEG) measures of evoked (EPs) and event-related potentials (ERPs) to assess speech perception in children with normal hearing [5,6,7] and in children who are hard-of-hearing [8,9,10,11]. One such measure is the mismatch response (MMR), which provides information about changes in brain activity that correspond to a change in various acoustic features (for a review see Näätänen et al. [12]). The MMR is likely generated bilaterally in the auditory cortex with contributions from the frontal cortex [13,14,15] and appears to be lateralized to the left hemisphere when processing speech [13,14]. The MMR is presumed to be an automatic, pre-perceptual change-detection response [12,15]. Developmental changes that occur over the first year of life are marked by decreased peak latencies, increased peak amplitudes, and a change in polarity from positive to negative in the MMR [5,16,17]. These changes appear to correspond with a child’s ability to process tones and increasingly complex auditory information such as speech [5,17,18,19]. Further, MMR has been correlated with behavioral measures of discrimination in young children [20] and with measures of later reading abilities [21]. Taken together, the MMR is an ideal measure of early speech discrimination processing and auditory perceptual development.

There is ample research suggesting that severe to profound hearing loss is related to delayed or abnormal auditory cortical development and that age at intervention is critical for children who receive cochlear implants [22,23,24,25,26]. In contrast, little is known about the impact that lesser degrees of hearing loss have on auditory development or the impact that reduced access to spoken language and impoverished signals have on the development of the central auditory system. For example, Koravand et al. [10] examined unaided EPs and ERPs in children ranging in age from 9–12 years with mild to moderate hearing loss, children with central auditory processing deficits, and children with normal hearing. They examined EP peak amplitudes (P1, N1, P2, and N2) and MMR amplitudes for speech (/ba-da/), simple nonspeech stimuli, and complex speech-like stimuli. Children with hearing loss had smaller N2 amplitudes compared to the children with normal hearing. MMR amplitudes did not differ between children with normal hearing and children who are hard of hearing. However, N1 and P2 responses were absent in almost 50% of the children in the hard-of-hearing cohort compared to 25% among the normal-hearing cohort. Smaller amplitudes could be the result of reduced audibility.

More recently, Calcus et al. [11] examined the MMR and another ERP, the late discriminative negativity (LDN), in response to speech and nonspeech sounds in two groups of children, younger children (*n* = 20; 8–12 years of age) and older children (*n* = 20; 12–16 years of age), with bilateral mild to moderate sensorineural hearing loss. Thirteen of the younger children had testing repeated a second time, six years later. Those children had present MMR responses to all contrasts when first measured but absent MMR responses to speech or speech-like stimuli after six years. However, the LDN did not change as a function of age. Those results were not attributed to changes in hearing thresholds or to group differences in audiometric thresholds, which may indicate changes in the auditory cortex measured by the MMR. Neither the Koravand et al. [10] nor Calcus et al. [11] studies assessed MMR while children were using HAs, which limits what we may infer about how children with hearing loss process speech sounds in their daily environment while wearing hearing aids (i.e., aided). These findings motivated our work to extend beyond unaided assessments of speech and to include aided assessments to evaluate HA fittings with the aim that, over time, it may be possible to alter intervention strategies as needed [27].

Our recent work [28,29,30] has focused on developing and refining an objective, MMR measure to assess speech perception of the neural correlates in response to changes in speech sounds in young infants with bilateral mild-to-severe permanent hearing losses recently fit with hearing aids. Using a time-frequency analysis of MMR responses in IHH (tested while wearing their hearing aids) and INH, we demonstrated a relationship between MMR in early infancy (1–6 months of age) and later behavioral measures of speech perception at 9–11 months of age [28]. We found that early brain responses (~100–300 ms after stimulation) in the 6–12 Hz frequency range, the alpha band, were positively correlated with later speech discrimination scores, assessed behaviorally. Those results corroborate other studies demonstrating the importance of the EEG alpha band for speech and language processing [31,32]. We hypothesized that early alpha band activity would reveal systematically larger response magnitudes when measured with HAs than without HAs and that such magnitude differences may serve as a candidate measure for validating HA fittings in young IHH.

## 2. Materials and Methods

### 2.1. Participants

Participants included 41 IHH aged 1.04 to 5.62 months (*M* = 3.34, *s.d*. = 1.04). Inclusion criteria for infants with mild to severe permanent bilateral hearing loss included: (1) hearing loss identified per the Colorado State Guidelines for Newborn Hearing Screening [33]; (2) fitted with bilateral HAs prior to 6 months of age; (3) currently used HAs; (4) were enrolled in early intervention; and (5) testing was completed for the same contrast in the aided (with HAs on) and unaided (with the HAs off) conditions; thus, each subject served as their own control. Due to poor EEG quality and/or multiple bad electrodes during testing three participants were excluded from the analysis. Table 1 provides a summary of the participant demographics and Appendix A provides details for each participant. Of the 38 retained participants, 20 were female and 18 were male. The better-ear pure tone average measured via auditory brainstem response testing (500, 1000, 2000, and 4000 Hz) ranged from 20 to 95 dB estimated hearing loss (*M* = 42.71 dBeHL, *s.d.* = 16.71 dB). All IHH wore bilateral air conduction HAs (for further details see Appendix A on hearing aid parameters). The duration of HA usage at the time of testing was 0.04 to 2.93 months (*M* = 1.10, *s.d*. = 0.76). Participant native language was English (*n* = 37) or Spanish (*n* = 1). Participants were given written consent by a parent to participate in the study as approved by the local Institutional Review Board.

### 2.2. EEG Procedure

Infants were placed in a rocker or held by a parent in a quiet, dim room to induce or aid sleeping during the test session. The rocker’s motion was not active during the EEG recordings. Eleven Ag/AgCl electrodes were placed on the scalp according to the International 10–20 system (F1, Fz, F2, C1, Cz, C2, P1, Pz, P2, M1, and M2) and were referenced to the nasion (Nz). An additional bi-polar recording channel was placed on the lateral canthus of the right eye and referenced to the superior orbit to monitor eye movement and waking. When a parent held the child, an additional ground electrode was placed on the parent’s forearm. Continuous EEG was recorded with a sampling rate of 1000 Hz and filtered from DC to 100 Hz during each experimental block using a Synamps2 EEG amplifier (Compumedics-Neuroscan, Charlotte, NC, USA).

### 2.3. Stimuli

Two different stimulus contrasts were used for testing MMR: a non-speech contrast consisted of a 500 Hz pure tone (PT) and a white noise burst (BBN), and a speech contrast consisted of/ba/(“bah”), and/da/(“dah”). Each non-speech stimulus was 500 ms in duration. The speech stimuli were natural speech tokens produced by a female speaker, digitized using a 16-bit analog-to-digital converter (AD Instruments Power Lab/16 SP) at 40 kHz and edited using Goldwave (Goldwave Inc., St. John’s, NL, Canada). For each speech stimulus, the sound was initiated by a consonant burst followed by the onset of voicing at 33 ms and formant transitions (F1 and F2) from 33 ms to 100 ms and followed by 400 ms of the steady-state vowel. Figure 1 shows a schematic illustration of the stimulus sequences used for testing. During testing, stimuli were presented in soundfield with a 1006 ms interstimulus interval (*SOA *= 1506 ms). This long interval increases the likelihood of identifying a cortical auditory evoked potentials response in young children [23,24]. Stimuli were presented at 70 dBA during testing. For additional details on stimulus creation, see Gilley et al. [29].

### 2.4. MMR Paradigm and Analysis Procedures

For MMR testing, the/PT/and/ba/sounds were treated as the standard stimulus, and the/BBN/and/da/sounds were treated as the deviant stimulus. Stimuli were presented in pseudo-random order at a ratio of 85% standard to 15% deviant, with the constraint that deviant stimuli could not appear more than twice in succession (see Figure 1). The order of testing conditions was balanced across subjects; some subjects were also tested with an additional contrast that was not analyzed for this study. Approximately 600 trials were collected for each MMR block.

### 2.5. EEG Signal Processing

EEG analysis included extraction of a latent, time-frequency representation of the auditory ERPs to characterize ERP activity in the alpha (~6–12 Hz) band (cf. Uhler et al. [28,30]). The basic procedure for EEG analysis was completed separately for each subject and amplification condition, and included the following five steps, described below (Section 2.5.1, Section 2.5.2, Section 2.5.3, Section 2.5.4 and Section 2.5.5).

#### 2.5.1. Band-Pass Filtering and Artifact Rejection

The continuous EEG data were filtered from 1 to 18 Hz (zero-phase, finite impulse response, −6 dB/octave). Regions of EEG containing transient artifact (e.g., spikes, myogenic potentials, external noise) or regions that occurred during non-sleep were marked as artifact and rejected from analysis. After artifact rejection, we retained an average of 377 total trials (*s.d. *= 61) and 56 deviant trials (*s.d. *= 13) per subject and condition, enough trials to observe an effect (cf. Leppänen et al. [34]). Note that for the purposes of computing averaged ERPs, the number of standard and deviant trials was matched for each subject (see Section 2.5.3, below; cf. [29,35]).

#### 2.5.2. Latent EEG Extraction

In a previous study, we demonstrated that the maximum ERP variance is captured in the first eigenvector of a spatial principal components analysis, and the channel loadings for that first eigenvector were maximally positive for the frontal-central electrodes Fz and Cz and maximally negative for the mastoid electrodes M1 and M2 [29]. In the present study, we retained the four electrodes (Fz, Cz, M1, and M2) and performed a principal components analysis of the spatial covariances matrix for these four channels. A latent representation of the continuous EEG was then extracted by simultaneously projecting the EEG data onto the first eigenvector (i.e., matrix dot product of the channel data and first eigenvector). This representation is functionally equivalent to computing a linked-mastoid reference, but with the advantage of discarding spatially discontinuous and noisy contributions to the experimental variance (cf. [29,36,37,38,39]). The result of this transform was a single, latent representation of the continuous EEG, which was then used for time-frequency analysis. We verified the latent representations by examining the directions of the channel loadings for each data block, which revealed that 76 of the 78 EEG representations had loadings in the expected directions (i.e., positive loadings for Cz and Fz; negative loadings for M1 and M2). The two blocks not revealing this pattern were from two separate subjects in the unaided condition only.

#### 2.5.3. ERPs and MMR

The continuous, latent EEG was segmented into epoched trials from −250 to 1250 ms around each stimulus onset and baseline corrected to the pre-stimulus interval separately for each trial. To compare standard and deviant responses and to compute the MMR responses, we matched the number of standard and deviant trials in each set. First, data were pooled into two sets of trials defined by whether a trial was standard or deviant and with the constraint that included trials must be both preceded by and followed by a standard trial. The number of trials in the standard pool was then reduced to match the number of trials in the deviant pool by selecting a random subset of the standard pool [29,35]. Averaged ERPs for the standard and deviant responses were computed as the mean of the retained trials for each type. The MMR was then computed by subtracting the standard ERP from the deviant ERP (i.e., deviant minus standard).

#### 2.5.4. Time-Frequency Coherence

A time-frequency representation of the continuous, latent EEG was extracted via the continuous wavelet transform with a 6-cycle Morlet wavelet and 200 log-spaced scales from 1 to 18 Hz. The transformed EEG was then segmented into epoched trials from −250 to 1250 ms around each stimulus onset and baseline corrected to the pre-stimulus interval separately for each trial and at each wavelet scale. The trials retained for averaging were the same as those used to compute the ERPs and MMR. Time-frequency coherence (TFC) maps were computed for the standard and deviant conditions in each block using a bootstrapped averaging procedure [28,29,40]. A bootstrapping procedure (*n*-boots = 10,009) was used to generate sets of surrogate TF maps by selecting a random permutation (with replacement) of each pool during each iteration and computing the mean. The complex-valued surrogate maps were normalized by dividing each point in each surrogate by its complex modulus. Separate standard and deviant TFC maps were then computed as the mean of all standard and deviant surrogates collapsed across stimulus types, respectively. The mean TFC maps were transformed into relative magnitudes by taking the squared absolute value.

#### 2.5.5. Alpha Band Separation

We extracted the mean alpha coherence magnitudes from the TFC maps. The exact frequency range for the alpha band was determined by computing the grand average TFC map for all subjects and all representations, and then computing the mean spectral envelope. The result of that procedure revealed alpha band activity in the range of 6 to 12 Hz, which corroborates our previous results [28]. The alpha coherence magnitudes were computed separately for each TFC map as the mean of all TFC scales within the respective bands defined by the group average. We interpret the coherence magnitudes as a measure of varying levels of alpha synchronization (greater coherence) and desynchronization (less coherence). The alpha coherence magnitudes for each ERP condition (standard and deviant) and each amplification condition (aided and unaided) were then treated as the dependent variables for statistical hypothesis testing.

To examine the influence of covarying factors on the observed alpha magnitudes, we computed the mean alpha magnitude for each subject in each listening condition and performed a series of comparisons for four co-factors: degree of hearing loss (*n* = 18 Mild: <40 dB hearing loss, *n* = 17 Moderate: <60 dB hearing loss, *n* = 4 excluded), age at test (*n* = 23 < 4 mo, *n* = 16 > 4 mo), age at HA fit (*n* = 28 < 3 mo, *n* = 11 > 3 mo), and sex (*n* = 20 Female, *n* = 18 Male). For each comparison we performed an unpaired, two-tailed *t*-test between alpha magnitudes for each sub-group.

### 2.6. Statistical Analyses and Hypothesis Testing

We sought to address three questions with these data: (1) What are the effects of HA amplification on detection of the auditory stimuli, (2) What are the effects of HA amplification on neural encoding of two competing contrasts, and (3) Does separation of the alpha band improve the detection of response features when compared to traditional ERP averaging? Additionally, we sought to determine whether alpha band differences were related to the aided SII, which provides a measure of the weighted proportion of the speech spectrum that is made audible by the HA and is related to improved outcomes among CHH. Separate statistical analyses were used to address each of these questions.

#### 2.6.1. Aided vs. Unaided Responses

To address our first question, we compared ERP amplitudes and alpha coherence magnitudes between the aided and unaided conditions separately for standard and deviant trials. Permutation *t*-tests were used for these comparisons and are described below (Section 2.6.5). Our hypothesis was that amplification would result in larger response magnitudes than with no amplification.

#### 2.6.2. Standard vs. Deviant Responses

Next, to address our second question, we compared ERP amplitudes and alpha response magnitudes between the standard and deviant trials separately for amplified/aided and unamplified/unaided responses. Permutation *t*-tests were used for these comparisons and are described below (Section 2.6.5). Our hypothesis was that the aided condition would result in larger coherence magnitudes than for the unaided condition.

#### 2.6.3. Alpha Responses vs. ERPs

To compare the performance of the alpha band separation to the traditional ERP averages, we computed the means and 95% confidence intervals of the measured response variable for each comparison group (standard and deviant, aided and unaided). Prior to averaging, the response waveforms for each subject were normalized to the RMS amplitude of the response, which allowed us to directly compare the alpha magnitudes with the ERP amplitudes. Means and confidence intervals were computed using a bootstrapped averaging procedure (random with replacement, *n*-boots = 10,009). We then compared the distribution of confidence interval ranges in each response epoch between the alpha responses and the ERP responses. Statistical testing was conducted with a parametric *t*-test (two-tailed, paired, FDR correction for multiple comparisons) of the confidence interval ranges for each comparison. Our hypothesis was that confidence interval ranges would be larger and more widely distributed for the ERP responses than for the alpha responses, which would indicate less group-level variability for the alpha responses.

#### 2.6.4. Aided Alpha vs. Aided SII

To examine the relationship between aided SII scores, which range from 0, meaning no audibility or access to speech sounds, to 1, meaning full audibility [41], we compared the peak alpha magnitudes for the aided deviant response to the better ear aided SII score. We then performed a Spearman rank correlation between these alpha-band MMR scores and the aided SII score. Our hypothesis was that there would be a significant correlation between the alpha magnitudes from the aided deviant response and the aided SII scores.

#### 2.6.5. Permutation *t*-Tests

Permutation *t*-tests (paired, two-tailed, *n*-permutations = 10,009) were performed separately at each time point for each comparison. For each test, data were divided into two initial groups for comparison (e.g., aided vs. unaided, or standard vs. deviant) and a paired, two-tailed *t*-test of each comparison produced the reference t-values for permutation testing. Next, a series of permutations was performed by randomly shuffling the group data and repeating the *t*-test for each permutation [42,43]. The largest t-value in each permutation was retained for the test distribution (Max-T method), which inherently corrects all statistical *p*-values for multiple comparisons by the family-wise error rate [44]. Statistical *p*-values were computed as the proportion of the test distribution with t-values greater than or equal to the reference t-values. Statistical significance was determined by *p*-values less than 0.05 and with the constraint that a significant value must be a member of a contiguous region of *p*-values < 0.05 with at least four members (i.e., a contiguous region ≥ 16 ms).

## 3. Results

### 3.1. Latent EEG Representation

Conversion of the raw EEG data to a latent representation revealed that the first eigenvector adequately captured the relevant ERP activity for all subjects. The percentage of variance accounted for by the first eigenvalue was computed separately for each subject and condition, and a paired, two-tailed *t*-test revealed no significant differences between the aided and unaided conditions for either contrast (Figure 2). These results suggest that each of the EEG blocks are represented by similar or approximately equal variances both within and between subjects and that experimental differences between these data are not due to unexplained differences from the transformation. A visual inspection of the group averaged ERP waveforms for each contrast and listening condition confirms the expected waveform shapes and morphologies (Figure 3).

### 3.2. ERPs and MMR

Figure 3 shows the group averaged ERP responses and 95% confidence intervals for each contrast and listening condition (panels a–d). Permutation *t*-tests revealed a significant difference (*p* < 0.05) between the standard and deviant responses only for the/PT-BBN/contrast and only in the aided listening condition in the time range of 0.28 to 0.40 s (280 to 400 ms; panel c). Figure 3 also shows the group averaged MMR waveforms (deviant minus standard) and 95% confidence intervals for each listening condition (panels e and f). There were no observable differences between the MMR waveforms for either condition.

### 3.3. Time-Frequency Representation

Figure 4a shows the grand averaged TFC map for all subjects and conditions, which corroborates our previous results [28,29,30]. The region representing the alpha band activity is denoted by the horizontal lines and labeled as “Alpha”. The grand averaged TFC also revealed a caveat to the selection of EEG bands defined by *a priori* frequency cutoffs, which is the presence of response activity at frequencies below the traditionally defined lower cutoff for the alpha band (~8 Hz). That finding also corroborates our previous descriptions of this activity, and we therefore included all activity in the 6–12 Hz range and refer to this as the alpha band. For the purposes of these analyses, we excluded the additional high theta-band activity that appeared at ~4–6 Hz for two reasons: (1) activity appeared at later latencies than the primary alpha responses, and (2) a preliminary examination of the high theta suggested that those responses were present even in cases when the alpha was dominant in the early portion of the response.

### 3.4. Cofactor Effects

Figure 4b shows boxplot distributions of the median and quartiles for each cofactor as a function of the alpha magnitudes collapsed across all trial conditions. A series of unpaired, two-tailed *t*-tests for each co-factor revealed no significant differences in alpha magnitude for any of the four co-factors (*p* > 0.05). Therefore, we presume that any observed effects in the alpha band are most likely attributed to experimental effects and not effects from one of the tested co-factors.

### 3.5. Temporal Envelopes of the Alpha Response

Figure 5a,b show the mean temporal envelopes and 95% confidence intervals for the selected frequency band and for each contrast (/ba-da/and/PT-BBN/) and include gray bars indicating time regions with significant differences from each permutation *t*-test (*p* < 0.05). Table 2 describes the mean peak magnitudes and peak latencies for each response. Peak response latencies were computed as the point of maximum amplitude within a contiguous region of significant differences.

#### 3.5.1. Comparison 1: Aided vs. Unaided Responses

Alpha coherence magnitudes were significantly greater in the aided than the unaided condition for the deviant/da/ (*p* < 0.05) and deviant/BBN/(*p* < 0.01) trials but were not different for the standard/ba/or standard/PT/trials (*p* > 0.05). Peak response latencies for the deviant/da/trials were observed at 188 ms and 156 ms for the aided and unaided responses, respectively. Peak response latencies for the deviant/BBN/trials were observed at 164 ms and 156 ms for the aided and unaided responses, respectively. These results suggest that the peak latencies for the aided responses occurred later than for the unaided responses.

#### 3.5.2. Comparison 2: Standard vs. Deviant Responses

Alpha coherence magnitudes for both the deviant/da/and deviant/BBN/trials were significantly greater than for the aided standard/ba/(*p* < 0.01) and aided standard/PT/(*p* < 0.05) trials, respectively. There was no significant difference in alpha responses for either speech contrast in the unaided conditions (*p* > 0.05). Peak response latencies of the aided alpha effects were observed at 180 ms for the/ba-da/contrast and at 152 ms for the/PT-BBN/contrast. Alpha responses from the/ba-da/speech contrast revealed two additional periods of significantly greater activity for the deviant/da/than for the standard/ba/trials in the aided condition, which appear much later in the response waveform (588 ms and 924 ms, *p* < 0.01).

### 3.6. Alpha Band Mismatch vs. Aided SII

Figure 6 shows boxplot distributions of the confidence interval ranges for each of the response groups. Results of the *t*-tests revealed a significant difference between the ERP and alpha distributions for all comparisons (*p* < 0.0001, FDR corrected). Specifically, the distribution of confidence interval ranges was smaller for the alpha responses than for the ERP responses, which suggests that alpha-band separation decreases the contribution of unwanted variability for group comparisons more than traditional ERP averaging.

### 3.7. Alpha Band Mismatch vs. Aided SII

Comparisons between the alpha-band deviant magnitudes and aided SII (see Figure 7) did not reveal a significant relationship (*r* = 0.04, *p* = 0.82), suggesting limitations in the utility of aided SII for explaining discrimination of fine acoustic differences such as /ba/versus/da/for our dataset. Notably, our dataset had limited variability in the calculated aided SII (see Appendix A), *(M* = 0.825, *sd* = 0.10, *range* = 0.45–0.95).

### 3.8. Post-Hoc Comparisons

Based on our results that alpha-band deviant responses were larger in the aided than unaided conditions; we sought to assess whether this effect may reflect the role of audibility for discrimination or whether such differences were driven by external energies generated by the HA processor. To control for HA processing, we compared the unaided responses as a function of hearing level (mild vs. moderate degrees of hearing loss). We hypothesized that if increased alpha-band magnitudes increased with audibility, we would observe differences between unaided infants with mild and moderate hearing loss such as those observed between the aided and unaided conditions. Data were divided into two groups based on the degree of hearing loss as determined by the four-frequency pure tone average, such that IHH with pure tone averages ≤ 40 dBHL were labeled as “Mild” (*n* = 17) and ≥41 dBHL and ≤60 dBHL were labeled as “Moderate” (*n* = 18). Four participants with pure tone averages > 60 dBHL were excluded from the post-hoc comparisons. Hypothesis testing was performed using a series of permutation *t*-tests (unpaired, two-tailed, *n*-permutations = 10,009) performed separately at each time point. For each test, an unpaired, two-tailed *t*-test produced the reference *t*-values for permutation testing. Next, a series of permutations was performed by randomly assigning new groups and repeating the *t*-test for each permutation [42,43]. The largest *t*-value in each permutation was retained for the test distribution (*Max-T* method), which corrects for multiple comparisons by the family-wise error rate. Statistical *p*-values were computed as the proportion of the test distribution with *t-values* greater than or equal to the reference *t*-values and statistical significance was determined using the same criteria described above for the aided vs. unaided comparisons.

#### Post-Hoc Results

Mean alpha coherence magnitudes as a function of hearing status are shown in Figure 8 and reveal a pattern of responses such as those described for the planned comparisons with the first prominent alpha peak appearing at approximately 100 to 250 ms after stimulus onset. Results of the permutation *t*-tests revealed no significant differences (*p* > 0.05) between mild and moderate hearing loss for the standard response in either the aided or unaided conditions. The deviant response in the aided condition revealed the largest alpha responses in both groups with a slightly later peak for the moderate hearing loss group; however, no statistical difference (*p* > 0.05) was observed for the early alpha peak. A significant difference (*p* < 0.05) was observed for a later peak in the waveform from 660 to 712 ms such that infants with moderate hearing loss show a large second peak while infants with mild hearing loss do not. This second alpha peak corresponds with the later peaks observed in the deviant response for the/ba-da/contrast (see Figure 5a). Finally, a significant difference (*p* < 0.05) was observed between deviant responses in the unaided condition from 144 to 196 ms such that alpha magnitudes are greater for mild hearing loss than moderate hearing loss. This difference observed in the deviant, unaided response supports our hypothesis that differences in alpha magnitude are likely driven by changes in audibility rather than epiphenomena of HA processing.

## 4. Discussion

To examine the effects of amplification on detection of the auditory stimuli we compared participants’ alpha responses for two different stimulus contrasts (/ba-da/and/PT-BBN/) with hearing aids in place (aided), and without hearing aids (unaided). Consistent with our previous work, we found that the first spatial eigenvector for both trial types accounted for most of the variance and provided a valid basis for examining auditory ERP activity. Also consistent with our previous findings, we found that a high-resolution time-frequency transformation of the EEG allowed consistent identification of frequency specific responses associated with detection of and discrimination between auditory stimuli. These analyses allowed us to describe brain activity corresponding to early auditory processing and to demonstrate the discrimination between two sounds for infants in the aided condition, but the same was not observed in the unaided condition.

### 4.1. Alpha Coherence Reveals Aided Benefit

When comparing aided and unaided responses, we found that amplification resulted in significantly greater alpha coherence magnitudes than responses without amplification, but only for the deviant trials. An important implication of this result is that auditory evoked potentials using only a single stimulus may not be useful for comparing aided and unaided responses or for demonstrating more subtle effects of stimulus discrimination possibly due to developmental differences in neural refractoriness and neural habituation [45]. For example, comparing responses from different stimuli that are presented in separate blocks of trials may reflect a different type of brain activity such as stimulus entrainment and may not be valid for drawing conclusions about how a listener can differentiate those sounds. Comparisons of the standard and deviant trials revealed that alpha coherence magnitudes for deviant trials were significantly greater than for standard trials in the aided condition but not in the unaided condition. That result suggests that in the aided condition the infant brains were able to differentiate the change between the standard and deviant stimuli for both contrast types. These results may have implications for understanding auditory development and may provide an objective measure for better demonstrating the importance of HA use in infants and young children.

### 4.2. Auditory Processing and the Alpha Band

Previous research has revealed conflicting results and interpretations of how synchronization and desynchronization of alpha activity contribute during different listening tasks. For example, Fujioka and Ross [46] observed alpha desynchronization in children aged 4–6 years while passively listening to noise bursts and violin sounds. These authors reported that desynchronization was more prominent for violin sounds than for noise bursts, and suggest developmental differences in hemispheric processing related to stimulus type as a possible explanation [46]. Stimulus relevance has also been demonstrated as a factor for alpha desynchronization in adults passively listening to their own name spoken by familiar voices when compared to unfamiliar voices del Giudice et al. [47].

In contrast, other studies have demonstrated increased alpha power or synchronization when cognitive demands such as working memory and attention are required. For example, Dyball et al. [48] reported increased alpha power during active listening, but not passive listening, when comparing responses to native and non-native speech tokens. Further, they showed that alpha synchronization was greater for native speech tokens, which seems to contradict the notion of stimulus relevance as a factor for driving desynchronization. Rather these findings suggest that language familiarity or experience may lead to larger alpha responses during active or effortful listening. Additionally, increased alpha synchronization has been observed in tasks requiring speech processing in noise, which is directly relevant to the processing demands of listeners with hearing loss. For example, Dimitrijevic et al. [32] reported increased alpha activity in normal hearing adults while actively attending to sequences of digits presented in noise but not during a passive listening task. Similarly, Paul et al. [31] reported greater alpha power in adult listeners with a cochlear implant during speech in noise tasks. Interestingly, the results of that study also revealed that alpha power decreased as listening reached maximal effort and was similar to low effort listening conditions; that is, they showed a non-linear “U-shaped” relationship between alpha modulation and listening effort. Paul et al. [31] suggest these findings may reflect the disengagement of task demands due to cognitive overload. Extended from those results, alpha activity may be a candidate measure for indexing the relative change in cognitive resources for auditory processing such as working memory and attention.

The results of this study likely reflect the development of early perceptual processes for distinguishing the importance or relevance in incoming sensory information. This interpretation may also help explain why differences were not observed between aided and unaided standard conditions. If the physiological mechanisms for gating or filtering of incoming information are involved in suppressing high probability or “low information” stimuli, then we should expect to observe less synchronization or lower alpha power as such responses become habituated. That is, we expect the brain to treat all low information stimuli in a similar manner. In this case, low information stimuli can either be repeated, high probability sounds or can be sounds that lack the spectral details (e.g., due to hearing loss) to differentiate new, low probability sounds such as those that occur with deviant trials. Such a mechanism would also lend support to our prior notion that these processes reflect the brain’s ability to distinguish novel or low probability sensory information as an internal measure of surprise [29].

### 4.3. Alpha Coherence vs. ERPs

A visual comparison of the 95% confidence intervals in Figure 2 and Figure 4 shows a clear difference in the morphology and variances of the waveforms between the averaged ERPs and the alpha coherence magnitudes. The difference in variances was confirmed by testing the distribution of confidence intervals for each response (Figure 6). Based on this result, we suggest that extracting frequency specific information from the EEG improves the detection and characterization of neural responses over traditional averaging methods. In our case, we chose the alpha band between 6 and 12 Hz based on results of a previous study and verified that choice by comparing the mean spectral envelopes from these data. Importantly, the selection of frequency specific bands should be based on some *a priori* knowledge or empirical computation of the spectral data. It is highly likely that a different experimental preparation could elicit responses in a different frequency band, which should also be considered for future studies of spectral EEG.

### 4.4. Alpha Coherence vs. Aided SII

Comparisons of the alpha magnitudes were not correlated with aided SII scores. The use of HAs in IHH is the primary method of assuring audibility of the auditory signal. Aided SII is a standard clinical tool for measuring how much speech-relevant information is accessible acoustically to children with hearing loss and is a hearing aid variable known to correlate with language outcomes in children [49,50]. For example, recent research suggests that children with aided SII values >0.71, auditory dosage (a measure of amount of time with auditory access), and word recognition in noise, could be used to predict spoken language abilities and guide changes in intervention [51]. Given the significant effects of alpha for the aided versus unaided responses and a lack of correlation with aided SII, the findings in this study suggest that aided SII alone may be inadequate for predicting hearing aid benefit in infants, especially when aided SII values are above 0.71. These findings provide further support for the need to develop and refine speech perception measures as a means of tracking HA benefit during infancy. While these findings were surprising, these results should be interpreted with caution because many infants in this study had high aided SII (i.e., >0.70), which may impact our ability to generalize these findings to children with poorer aided SII. As with previous studies, even among children with high aided SII values, there continues to be variability in outcome measures such as parent report of auditory development [52] and speech discrimination [53]. Taken together, these findings support the need to use a diagnostic battery including aided SII and speech perception as is used among older children and adults.

### 4.5. Limitations and Future Directions

While this study provides insight to the benefits of amplification for early perceptual development in IHH, there are several limitations that should be considered when interpreting these results. Time is limited to the average sleep cycle in young infants, approximately 50–60 min [54], which limits the number of stimuli we can present in a single EEG session. Our preference would have been to record two different auditory contrasts (i.e.,/ba-da/and/PPT-BBN/) within the same group of infants in both the aided and unaided condition. For example, a broad-band noise burst inherently contains more energy than a tone burst, and we may deduce that the observed differences were simply a reflection of these energy differences. However, given that the unaided responses did not reflect this energy difference, our conclusion remains that these results demonstrate the benefits of amplification for discriminating acoustic differences. Future studies focused on within-group comparisons can help to differentiate these effects.

We used a paradigm similar to others studying older children who are hard-of-hearing, and who did not interchange the standards and deviants [10,11]. While replication of this aspect of their study design is a weakness, we have no reason to believe that these results, or the interpretation of these results are affected by this limitation. Future research will need to consider the effects of different stimulus paradigms and measure the output of the stimuli after being processed by the HA in the infants’ ear canal.

## 5. Conclusions

We measured both cortical auditory evoked potentials and ERPs in the aided and unaided conditions. While our work aims to extend beyond the detection of sound into speech perception of the neural correlates in response to changes in speech sounds, the latter would not be possible without first detecting a sound. ERP measures can be used to examine both detection and discrimination of speech and non-speech sounds in young sleeping IHH. Significant differences were observed in the alpha-band response to the deviant stimulus in the aided condition. No differences were observed between standard and deviant responses in the unaided condition for the speech contrast. Additionally, no relationship was observed between the alpha-band MMR and aided SII. These findings suggest the utilization of ERPs, in addition to HA verification, could be used to assess speech discrimination. Thus, over time and replication of results, ERP utilization could expand and refine our abilities to validate HA fittings among IHH.

## Figures and Tables

**Figure 1 children-10-00600-f001:**
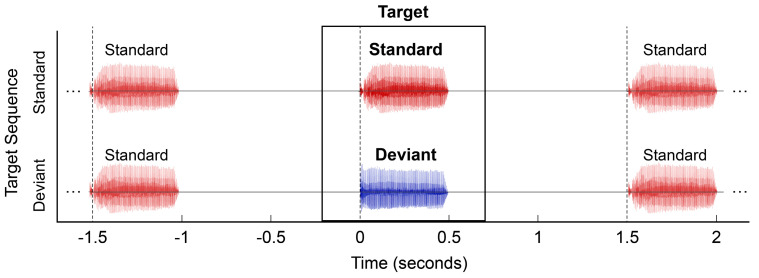
Schematic illustration of the stimulus sequences used for testing. Each waveform represents the presentation of either a “Standard” sound (/ba/or/PT/) shown in red or of a “Deviant” sound (/da/or/BBN/) shown in blue. The abscissa represents time (in seconds) relative to the onset of an identified target stimulus in the presentation sequence.

**Figure 2 children-10-00600-f002:**
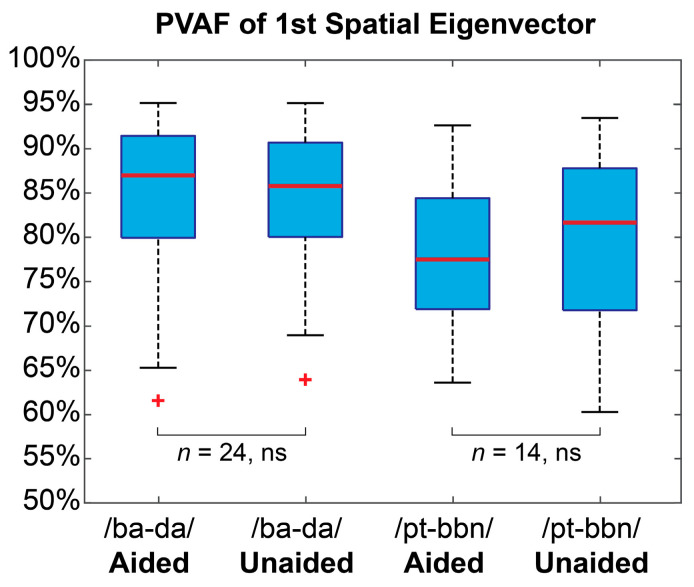
**Spatial eigenvector variances.** Boxplots represent distributions for the percentage of variance accounted for (PVAF) by the first spatial eigenvector of four EEG channels (Fz, Cz, M1, and M2) for the aided and unaided conditions and for each contrast. Statistical comparisons between the aided and unaided conditions were computed separately for each contrast (*ns* = not significant, *p* > 0.05). + signifies points that are outside of the interquartile range.

**Figure 3 children-10-00600-f003:**
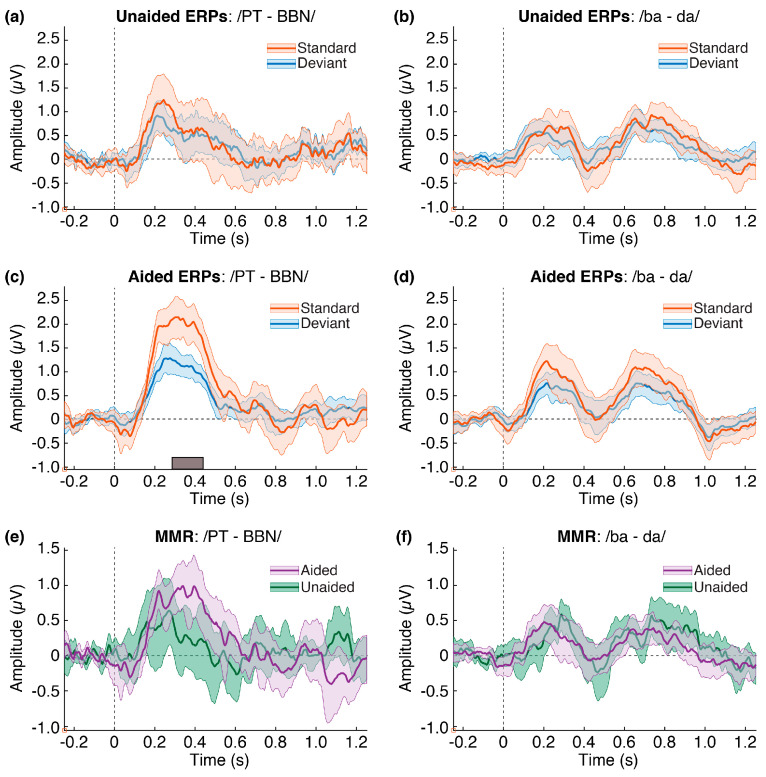
**Grand averaged ERP responses (a–d) and MMR (e,f) for each contrast and listening condition.** In each panel, time (seconds), is represented on the abscissa with stimulus onset at time = 0 and relative amplitude (microvolts) is represented on the ordinate. Solid lines in each waveform represent the group averaged mean response and shaded regions represent the 95% confidence intervals for 10,009 bootstrapped permutations of the subject responses.

**Figure 4 children-10-00600-f004:**
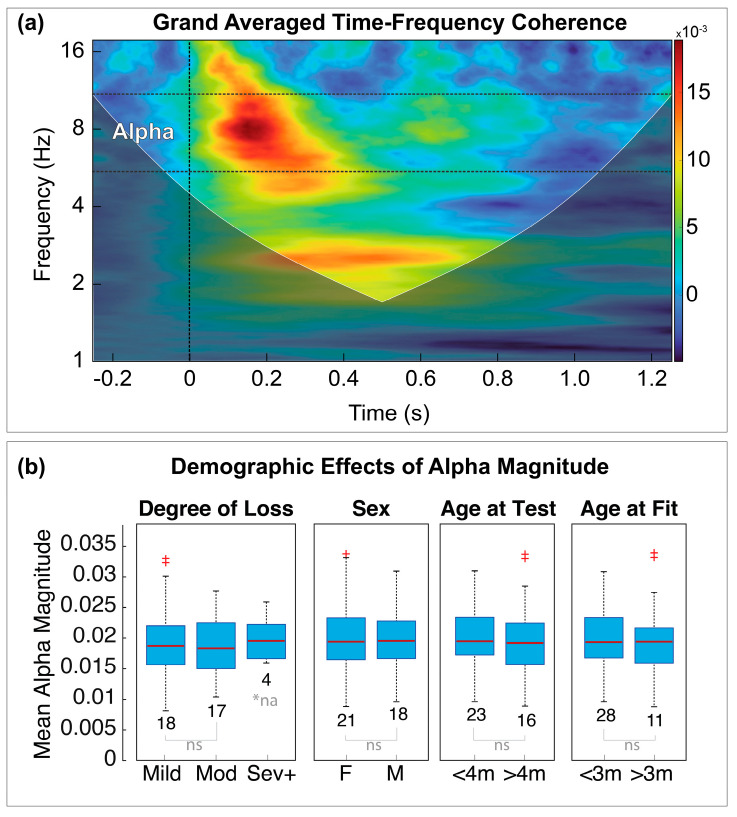
**Time-frequency coherence and alpha band co-factor effects.** (**a**) Grand averaged time-frequency coherence map of all conditions and contrasts representing the region of activation retained for the alpha band ERPs (6–12 Hz, represented by the horizontal, dotted lines). The vertical dotted line at *time* = 0 s represents the relative time of onset for each stimulus. The magnitude of coherence relative to the pre-stimulus baseline is represented by the color scale. The dark, shaded region in the lower portion of the figure represents time-frequency points outside of the cone-of-influence for this epoch window. (**b**) Boxplots of the distributions for mean alpha magnitudes collapsed across all trial types (standards, deviants, and contrasts) as a function of four different co-factors: *Degree of loss*, *Sex*, *Age at test*, and *Age at (HA) fit*. Statistical comparisons between each sub-group were computed separately for each co-factor (*ns* = not significant, *p* > 0.05), *na = not assessed). + signifies points that are outside of the interquartile range. For the following co-factors: degree of hearing loss, age at test, and age at hearing aid fit two points that were outside of the interquartile range, represented by two ‘+’ stacked vertically.

**Figure 5 children-10-00600-f005:**
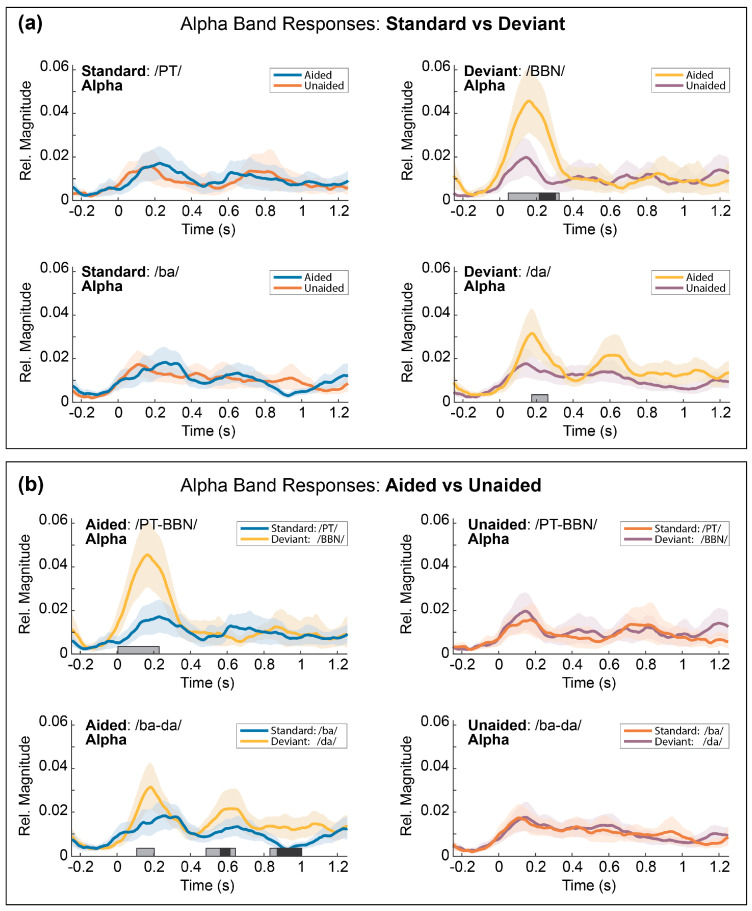
**Mean temporal envelopes of alpha band coherence: planned comparisons**. (**a**) Mean temporal envelopes for the alpha band coherences showing comparisons of the standard versus deviant responses for each contrast and each listening condition. Responses for the/PT-BBN/contrast are shown in the upper row and responses for the/ba-da/contrast are shown in the bottom row. Responses are shown for the aided condition (left column) and responses for the unaided condition (right column). The shaded regions around each temporal envelope represent the 95% confidence intervals for the group mean and the shaded bars along the abscissa represent time regions with significant differences between the aided and unaided responses (gray = *p* < 0.05, black = *p* < 0.01). (**b**) Mean temporal envelopes for the alpha band coherences showing comparisons of the aided versus unaided responses for each stimulus. Responses for the/PT-BBN/contrast are shown in the upper row and responses for the/ba-da/contrast are shown in the bottom row. Responses for the standard stimulus of each contrast are shown in the left column and responses for the deviant stimulus for each contrast are shown in the right column. The shaded regions around each temporal envelope represent the 95% confidence intervals for the group mean and the shaded bars along the abscissa represent time regions with significant differences between the aided and unaided responses (gray = *p* < 0.05, black = *p* < 0.01).

**Figure 6 children-10-00600-f006:**
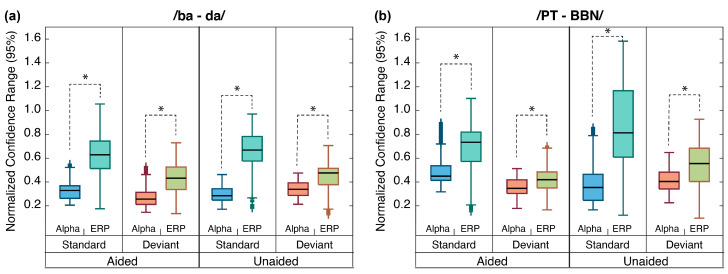
**Alpha band vs. ERP variability.** Boxplots represent the range of the 95% confidence intervals for each group-averaged response for each stimulus contrast: (**a**) /ba-da/ and (**b**) /PT-BBN/. Boxes in each panel are grouped according to each Alpha vs. ERP comparison for the standard and deviant responses and for the aided and unaided responses. The asterisk (*) above each comparison set indicates a significant difference (*p* < 0.001, FDR corrected).

**Figure 7 children-10-00600-f007:**
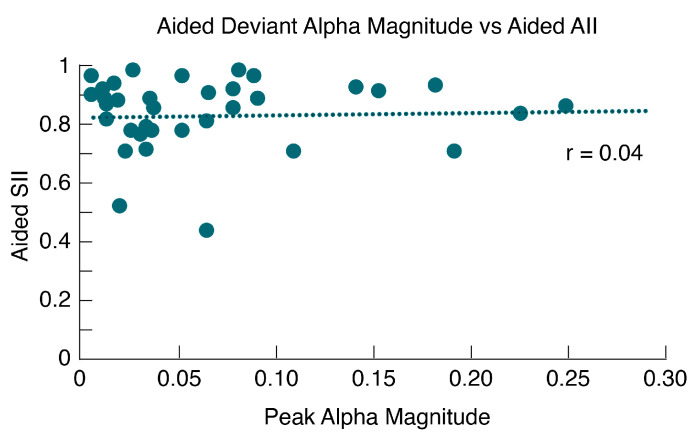
**Alpha magnitude as a function of aided speech intelligibility (SII)**. Alpha magnitudes for the deviant response in the unaided condition are represented along the abscissa and aided SII scores are represented along the ordinate. A correlation analysis revealed no significant correlation between alpha magnitude and aided SII (*r* = 0.04, *p* > 0.05).

**Figure 8 children-10-00600-f008:**
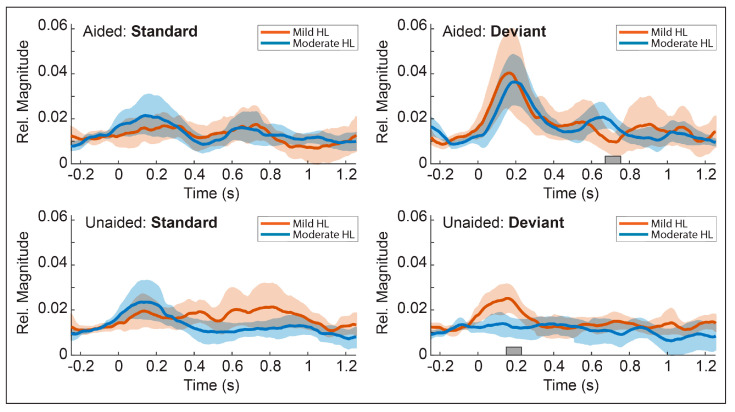
**Mean temporal envelopes of alpha band coherence: post-hoc comparisons.** Mean temporal envelopes for the alpha band coherences showing comparisons of infants with Mild hearing loss versus Moderate hearing loss collapsed across contrast type. Responses for the aided condition are shown along the top row and responses for the unaided condition are shown along the bottom row. Standard responses are shown in the left column and deviant responses in the right column. Shaded bars along the abscissa represent time regions with significant differences between the Mild and Moderate hearing loss groups (gray = *p* < 0.05).

**Table 1 children-10-00600-t001:** Summary of participant demographics.

Total N = 38 (20 Female)	*Mean (s.d.)*	Range
Age at test	3.84 (1.16) months	1.75–6.32 months
Age at hearing aid fit	2.74 (0.99) months	1.12–6.17 months
Duration of hearing aid use	1.10 (0.76) months	0.04–2.93 months
**Degree of hearing loss (PTA range)**	**Count**	
Mild (15–39 dB HL)	*n* = 21	
Moderate (40–59 dB HL)	*n* = 12	
Moderately-severe (≥60 dB HL)	*n* = 5	
**ERP Contrast**	**Count**	
/PT-BBN/	*n* = 14	
/ba-da/	*n* = 24	

Note: Age at ERP assessment, age when hearing aids were first fit, and duration of hearing aid use when the ERP assessment was conducted. PTA = pure tone average for 500, 1000, 2000, and 4000 Hz and is representative of the child’s better hearing ear. Each infant was only assessed using one ERP contrast in both the aided (with hearing aids on) and unaided (without hearing aids on) conditions. Individual participant demographics can be found in Appendix A.

**Table 2 children-10-00600-t002:** Alpha band response comparisons.

Contrast	Comparison	Aided Magnitude (*CI 95%)*	Unaided Magnitude (*CI 95%)*	Effect	Latency (ms)
/PT-BBN/	Aided vs. Unaided Deviant	0.046 (0.031,0.060)	0.016 (0.011,0.028)	*p* < 0.05	A: 164 ms, U: 156 ms
/PT-BBN/	Aided vs. Unaided Standard	0.017 (0.009,0.025)	0.015 (0.008,0.023)	ns	
/ba-da/	Aided vs. Unaided Deviant	0.031 (0.018,0.041)	0.018 (0.010,0.025)	*p* < 0.01	A: 188 ms, U: 156 ms
/ba-da/	Aided vs. Unaided Standard	0.018 (0.013,0.025)	0.017 (0.012,0.023)	ns	
**Condition**	**Comparison**	**Deviant magnitude (*CI 95%)***	**Standard magnitude (*CI 95%)***	**Effect**	**Latency (ms)**
Aided	Deviant/BBN/vs. Standard/PT/	0.046 (0.031,0.060)	0.017 (0.009,0.025)	*p* < 0.05	152 ms
Unaided	Deviant/BBN/vs. Standard/PT/	0.016 (0.011,0.028)	0.015 (0.008,0.023)	ns	
Aided	Deviant/da/vs. Standard/ba/	0.031 (0.018,0.041)	0.015 (0.009,0.021)	*p* < 0.05	180 ms
Unaided	Deviant/da/vs. Standard/ba/	0.018 (0.010,0.025)	0.017 (0.010,0.023)	ns	

Note: Mean peak magnitudes and peak latencies and *95% confidence intervals (CI)* by frequency band, contrast (/ba-da/and/PT-BBN/) and condition (Aided and Unaided). Time regions with significant differences from each permutation *t*-test are reported in milliseconds (ms) from the stimulus onset. Time regions which are not significant (ns) are indicated accordingly.

## Data Availability

De-identified data will be made available after institutional review and approval.

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
