# Peer review of "EEG Alpha Band Responses Reveal Amplification Benefits in Infants with Hearing Loss"

_children, 2023, doi:10.3390/children10030600_

Round 1

Reviewer 1 Report

The manuscript describes a study of the physiological mismatch response (MMR) in aided and unaided conditions, in infants who were hard-of-hearing (IHH).  The MMR offers some advantages over passive physiological measures, for infants whose ability to provide behavioral responses is limited.  The well-designed and technically-sophisticated study addressed a clinically-important issue.  The results are presented in appropriate detail, and the Discussion is informative, placing the results in context and considering both the strengths and limitations of the project.

Author Response

We are grateful for your feedback and appreciate your thoughtful and thorough review. 

Reviewer 2 Report

This study examined the effects of hearing aid amplification on auditory detection and discrimination in infants who were hard-of-hearing (IHH). Results showed that aided alpha band responses were greater than unaided responses for deviant trials, but not for standard trials. This suggests that the alpha band mismatch can be used to examine both detection and discrimination of speech and non-speech sounds in IHH.

There are several theoretical and technical issues regarding the rationale, experimental design, data analysis and interpretation that need to be addressed.

1. The literature review needs to cover maturation of the evoked responses and the mismatch response (MMR) for speech and nonspeech sounds in early infancy. In addition, there needs to be literature review of the target measure of alpha-band activity in MMR and its functional interpretation. Without a comprehensive review, the rationale for the current design appears inadequate. For instance, it is unjustified why one condition used pure tone vs. white noise and another condition used /ba/ vs/ /da/ for testing the infants between 1 and 6 months of age. What did previous studies use to test infants of this age range? What were the findings?

2. There is a lack of control group. No data from a control group of typically developing infants in the same age range who do not have hearing loss were available for comparison. For instance, it is unclear whether the ERP responses in the aided condition would be similar to those obtained from normal-hearing infants.

3. There are fundamental flaws in the stimulus choice and serious confounds in the data analysis. The acoustic properties of pure tone and white noise were entirely different. Why did the researchers choose to use tone vs. noise for an MMR study? Because the standard stimuli with 85% occurrence had a many more trials than the deviant at 15%, the disproportional averaging of ERP trials and the acoustic differences between them would create differences in the ERP waveforms regardless of whether MMR was actually elicited or not. The researchers need to first establish that significant MMR activities were elicited by controlling the confounding variables of a huge trial number difference (377 vs. 56 as reported after artifact removal) and acoustic differences before making comparisons between the aided and unaided conditions. Even without the oddball paradigm, /ba/ and /da/ ERP waveforms would show differences. The averages of 377 /ba/'s and 57 /ba/'s would also show large differences.

4. It seems unjustified that the researchers did not report the MMR results (amplitude and latency data) directly. Instead, the alpha-band activity was chosen as the focus of findings. Why? Could the alpha activity simply reflect the rhythm of the stimulus presentation protocol with an SOA of 1506 ms (equivalent to 6~7 Hz)?

5. The quantification of the alpha activity did not follow previous studies. I do not understand why the researchers came up with their own method of EEG quantification for the alpha activity using 4 chosen electrodes with a principal component analysis approach. No one else in the EEG field used such a method to my knowledge.

Author Response

  1. The literature review needs to cover maturation of the evoked responses and the mismatch response (MMR) for speech and nonspeech sounds in early infancy. In addition, there needs to be literature review of the target measure of alpha-band activity in MMR and its functional interpretation. Without a comprehensive review, the rationale for the current design appears inadequate. For instance, it is unjustified why one condition used pure tone vs. white noise and another condition used /ba/ vs/ /da/ for testing the infants between 1 and 6 months of age. What did previous studies use to test infants of this age range? What were the findings?

Thank you for this suggestion, we tried to keep our literature review brief and succinct allowing focus on the need to develop a tool to validate hearing aid fittings while simultaneously describing what is known and not know among children with hearing loss. We have added a high level overview of how MMR changes over the first few months of life, now in lines 48-54 to supplement our existing literature review. I think this will help the reviewer understand our approach in this line of inquiry and why we focused on such a narrow age range.

  1. There is a lack of control group. No data from a control group of typically developing infants in the same age range who do not have hearing loss were available for comparison. For instance, it is unclear whether the ERP responses in the aided condition would be similar to those obtained from normal-hearing infants.

In previous studies, we compared MMR responses from infants with hearing loss and infants with typical development and have demonstrated that the response characteristics are quite similar, especially in infants with mild or moderate hearing loss. In the present study, however, our question involved within-subject changes specifically related to the use of hearing aids; that is, each subject served as their own control. Therefore, the response characteristics of typically developing infants were not relevant to whether a change occurred with amplification. In general, we felt that including a typically developing cohort would add unnecessary complexity to the study and would detract from the comparison of interest, here (i.e., aided vs unaided responses).

  1. There are fundamental flaws in the stimulus choice and serious confounds in the data analysis. The acoustic properties of pure tone and white noise were entirely different. Why did the researchers choose to use tone vs. noise for an MMR study?

We chose a tone vs noise contrast for two reasons: 1) this contrast has been used to demonstrate infant discrimination in previous studies by other research groups (e.g., Kushnerenko et al. [1]). When initially designing this study, we felt that it was important to demonstrate replication of the MMR results between those previous studies and our own.  2) The stark contrast between a tone and a noise burst provides a demonstration of MMR sensitivity; that is, by comparing this very large acoustic difference to a smaller, more discrete difference (e.g, /ba/ vs /da/), we observe how this measure informs us about discrimination processing. We provide a more detailed description of the similarities and differences of these contrasts (non-speech vs speech) in our previous research (e.g., Gilley et al., 2017, Uhler et al., 2019).

Because the standard stimuli with 85% occurrence had a many more trials than the deviant at 15%, the disproportional averaging of ERP trials and the acoustic differences between them would create differences in the ERP waveforms regardless of whether MMR was actually elicited or not. The researchers need to first establish that significant MMR activities were elicited by controlling the confounding variables of a huge trial number difference (377 vs. 56 as reported after artifact removal) and acoustic differences before making comparisons between the aided and unaided conditions. Even without the oddball paradigm, /ba/ and /da/ ERP waveforms would show differences. The averages of 377 /ba/'s and 57 /ba/'s would also show large differences.

We apologize to the reviewer for the confusion here. We did, in fact, correct for the discrepancy in trial counts, but that procedure was described in a later section (2.5.3). We have amended this section of the methods to improve the clarity of that procedure.

  1. It seems unjustified that the researchers did not report the MMR results (amplitude and latency data) directly. Instead, the alpha-band activity was chosen as the focus of findings. Why?

Our choice of the alpha band was based on a priori evidence. We state in the introduction:

“We found that early brain responses (~100-300 ms after stimulation) in the 6 – 12 Hz frequency range, the alpha band, were positively correlated with later speech discrimination scores, assessed behaviorally. Those results corroborate other studies demonstrating the importance of the EEG alpha band for speech and language processing [1],[2]. (Page 2; Lines 89-93).

Further, our inclusion of the standard MMR waveforms (Figure 2) clearly shows very little difference for this experimental effect. Given that our observations in the alpha band are explanatory while the traditional ERPs are not, we did not feel it was necessary to perform further analyses when a significant effect is not observed.

Could the alpha activity simply reflect the rhythm of the stimulus presentation protocol with an SOA of 1506 ms (equivalent to 6~7 Hz)?

The effects of stimulus entrainment most certainly could influence these observations. In our case we used a very long SOA, which would be observed as entrainment at frequencies less than 1 Hz (an SOA of 1.506 seconds is an entrainment frequency of 0.66 Hz, not 6~7 Hz as suggested). Further, even if we assume that the responses are driven by entrainment mechanisms, the differences in the expression of such entrainment are still explained by the experimental effect (i.e., no amplification vs amplification).

  1. The quantification of the alpha activity did not follow previous studies. I do not understand why the researchers came up with their own method of EEG quantification for the alpha activity using 4 chosen electrodes with a principal component analysis approach. No one else in the EEG field used such a method to my knowledge.

Perhaps I’m (PG) misunderstanding the reviewer’s comment, here. The use of both PCA and time-frequency (TF) transforms for EEG and ERP analyses is not a new method of quantification [2]–[16]. Rather, PCA and TF have been ubiquitous tools for decades and are included as standard analysis transforms even in commercial and clinical applications (e.g., Compumedics Neuroscan, Brain Vision Analyzer, etc.). The availability of analysis tools such as the “ERP PCA Toolkit” for MATLAB further demonstrate the widespread use of these applications [10]. More specifically, spatiotemporal reduction of the EEG was demonstrated for use in an oddball paradigm as early as 2001 [17] and more recently for infants by our group [18]. Similar methods have also been used in studies of similar clinical populations [15], [16], [19], [20]. Finally, we would refer to the consensus papers that specifically address the applications of PCA [4], [8], [10]; for example, Kayser and Tenke stated in 2006, “We are in total accord that PCA affords superior description and quantification of ERP components when compared to other conventional methods (e.g., peak or window estimates).”

In the present study, our choice of the 4 chosen electrodes (Cz, Fz, M1, and M2) was based on a priori evidence that those electrodes provide the best representation of the auditory ERPs in this population [18], [21]. Similarly, our choice of the alpha band was also based on our previous research (Please see our response to the question (above) regarding our selection of the alpha band).

We have added the relevant references to support this methodology to the description of latent waveform extraction (Section 2.5.2.).

[1]        E. Kushnerenko et al., “Maturation of the auditory event-related potentials during the first year of life,” NeuroReport, vol. 13, no. 1, pp. 47–51, Jan. 2002, doi: 10.1097/00001756-200201210-00014.

[2]        J. Kayser and C. E. Tenke, “Optimizing PCA methodology for ERP component identification and measurement: theoretical rationale and empirical evaluation,” Clin. Neurophysiol., vol. 114, no. 12, pp. 2307–2325, Dec. 2003, doi: 10.1016/S1388-2457(03)00241-4.

[3]        J. Kayser and C. E. Tenke, “Principal components analysis of Laplacian waveforms as a generic method for identifying ERP generator patterns: II. Adequacy of low-density estimates,” Clin. Neurophysiol., vol. 117, no. 2, pp. 369–380, Feb. 2006, doi: 10.1016/j.clinph.2005.08.033.

[4]        J. Kayser and C. E. Tenke, “Consensus on PCA for ERP data, and sensibility of unrestricted solutions,” Clin. Neurophysiol., vol. 117, no. 3, pp. 703–707, Mar. 2006, doi: 10.1016/j.clinph.2005.11.015.

[5]        C. Tenke and J. Kayser, “Reference-free quantification of EEG spectra: Combining current source density (CSD) and frequency principal components analysis (fPCA),” Clin. Neurophysiol., vol. 116, no. 12, pp. 2826–2846, Dec. 2005, doi: 10.1016/j.clinph.2005.08.007.

[6]        C. E. Tenke and J. Kayser, “Surface Laplacians (SL) and phase properties of EEG rhythms: Simulated generators in a volume-conduction model,” Int. J. Psychophysiol., vol. 97, no. 3, pp. 285–298, Sep. 2015, doi: 10.1016/j.ijpsycho.2015.05.008.

[7]        F. Scharf and S. Nestler, “Principles behind variance misallocation in temporal exploratory factor analysis for ERP data: Insights from an inter-factor covariance decomposition,” Int. J. Psychophysiol., vol. 128, pp. 119–136, Jun. 2018, doi: 10.1016/j.ijpsycho.2018.03.019.

[8]        F. Scharf, A. Widmann, C. Bonmassar, and N. Wetzel, “A tutorial on the use of temporal principal component analysis in developmental ERP research – Opportunities and challenges,” Dev. Cogn. Neurosci., vol. 54, p. 101072, Apr. 2022, doi: 10.1016/j.dcn.2022.101072.

[9]        J. Dien, W. Khoe, and G. R. Mangun, “Evaluation of PCA and ICA of simulated ERPs: Promax vs. infomax rotations,” Hum. Brain Mapp., vol. 28, no. 8, pp. 742–763, 2007, doi: https://doi.org/10.1002/hbm.20304.

[10]      J. Dien, “The ERP PCA Toolkit: An open source program for advanced statistical analysis of event-related potential data,” J. Neurosci. Methods, vol. 187, no. 1, pp. 138–145, Mar. 2010, doi: 10.1016/j.jneumeth.2009.12.009.

[11]      E. M. Bernat, W. J. Williams, and W. J. Gehring, “Decomposing ERP time–frequency energy using PCA,” Clin. Neurophysiol., vol. 116, no. 6, pp. 1314–1334, Jun. 2005, doi: 10.1016/j.clinph.2005.01.019.

[12]      E. M. Bernat, S. M. Malone, W. J. Williams, C. J. Patrick, and W. G. Iacono, “Decomposing delta, theta, and alpha time–frequency ERP activity from a visual oddball task using PCA,” Int. J. Psychophysiol., vol. 64, no. 1, pp. 62–74, Apr. 2007, doi: 10.1016/j.ijpsycho.2006.07.015.

[13]      E. Edwards et al., “Comparison of time-frequency responses and the event-related potential to auditory speech stimuli in human cortex.,” J. Neurophysiol., vol. 102, no. 1, pp. 377–386, 2009, doi: 10.1152/jn.90954.2008.

[14]      C. S. Herrmann, S. Rach, J. Vosskuhl, and D. Str??ber, “Time-frequency analysis of event-related potentials: A brief tutorial,” Brain Topogr., vol. 27, no. 4, pp. 438–450, 2014, doi: 10.1007/s10548-013-0327-5.

[15]      P. M. Gilley, A. Sharma, M. Dorman, C. C. Finley, A. S. Panch, and K. Martin, “Minimization of cochlear implant stimulus artifact in cortical auditory evoked potentials,” Clin. Neurophysiol., vol. 117, no. 8, pp. 1772–1782, 2006, doi: 10.1016/j.clinph.2006.04.018.

[16]      P. M. Gilley and A. Sharma, “Functional Brain Dynamics of Evoked and Event-Related Potentials From The Central Auditory System,” Perspect. Hear. Hear. Disord. Res. Diagn., 2010, doi: 10.1044/hhd14.1.12.

[17]      K. M. SPENCER, J. DIEN, and E. DONCHIN, “Spatiotemporal analysis of the late ERP responses to deviant stimuli,” Psychophysiology, vol. 38, no. 2, pp. 343–358, 2001, doi: 10.1111/1469-8986.3820343.

[18]      P. M. Gilley, K. Uhler, K. Watson, and C. Yoshinaga-Itano, “Spectral-temporal EEG dynamics of speech discrimination processing in infants during sleep,” BMC Neurosci., vol. 18, no. 1, 2017, doi: 10.1186/s12868-017-0353-4.

[19]      P. M. Gilley, N. K. Walker, and A. Sharma, “Abnormal oscillatory neural coupling in children with language-learning problems and auditory processing disorder,” Semin. Hear., vol. 35, no. 1, pp. 15–26, 2014, doi: 10.1055/s-0033-1363521.

[20]      P. M. Gilley, A. Sharma, and M. F. Dorman, “Cortical reorganization in children with cochlear implants,” Brain Res., vol. 1239, pp. 56–65, 2008, doi: 10.1016/j.brainres.2008.08.026.

[21]      K. M. Uhler, S. K. Hunter, E. Tierney, and P. M. Gilley, “The relationship between mismatch response and the acoustic change complex in normal hearing infants,” Clin. Neurophysiol., vol. 129, no. 6, pp. 1148–1160, Jun. 2018, doi: 10.1016/j.clinph.2018.02.132.

Reviewer 3 Report

Appraisal:

“EEG alpha band responses reveal amplification benefits in infants with hearing loss”

Introduction: In the introduction the authors allow, not only the epidemiology of IHH in US, but the principal methodology used to diagnosis and results obtained in previous published studies. According to me, this is very important to guide the reader, also who have not expertise (clinical or scientific) in IHH. Indeed, the authors performed a critical review of previous findings, resulting in the aims and hypotheses presented at the end of introduction.

Methods: The methods are well-written and the authors explained in detail all the phases and statistical tests that they used. Authors can find some suggestions as follow:

Please add a reference for “Colorado State Guidelines for Newborn Hearing Screening”.

Authors did not mention the presence of HC group. Please, explain this methodological choice.

“Participant native language was English or Spanish”Why did the authors selected 2 different groups? Can be considered a limitation? Please add to table 1 the n. of Spanish vs English native speakers.

Stimuli: I advise to add a schematic figure.

Did the authors performed an ICA?

2.6. Statistical Analyses and Hypothesis Testing- I suppose that this paragraph can cause some confusion in the readers. I suggest to organize this section in 2 parts preprocessing and Statistical Analyses and Hypothesis Testing with sub-paragraphs. It is a suggestion.

I do not agree with the permutation t-test. It is correct, but I advise the authors to motivate the use of permutation t-test instead of different statistical tests.

Despite their complexity, the results are interesting and follow the hypotheses as explained by the authors in the methods section.

Paragraph 3.6- p < 0.0001, FDR corrected. Please, check the value of p.

Discussion: Discussion should start with a summary of the experiment and highlights the principal aspects of novelty, after that I agree with the subparagraphs. BTW, I found it very interesting.

Author Response

Please add a reference for “Colorado State Guidelines for Newborn Hearing Screening”.

Thank you, this reference has now been added to the methods section.

Authors did not mention the presence of HC group. Please, explain this methodological choice.

We address the use of a control group in response to reviewer 2 (above).

“Participant native language was English or Spanish”Why did the authors selected 2 different groups? Can be considered a limitation? Please add to table 1 the n. of Spanish vs English native speakers.

In this study we did not consider these children as two separate groups because, 1) all speech sounds are considered native sounds in both English and Spanish, and 2) our goal of assessing changes from amplification are not dependent on nor predictive of language-specific development.

We do, however, agree that this information should be included in the participant description and have specified that information in the text of section 2.1 (English n = 37; Spanish n = 1).

Stimuli: I advise to add a schematic figure.

          We have now added a schematic figure of the stimuli as Figure 1.

Did the authors performed an ICA?

No, in this case we used a PCA-only approach.

2.6. Statistical Analyses and Hypothesis Testing- I suppose that this paragraph can cause some confusion in the readers. I suggest to organize this section in 2 parts preprocessing and Statistical Analyses and Hypothesis Testing with sub-paragraphs. It is a suggestion.

Indeed, given the complexity of these methods we would like to make everything as clear as possible. In this case, we are struggling to understand the reviewer’s suggestion for changes, but would be more than happy to accommodate if we can receive further/more detailed feedback (e.g., how might we order the sections better to separate the sub-paragraphs, or which information is most confusing that we should focus on, etc.)? Thank you for the suggestions.

I do not agree with the permutation t-test. It is correct, but I advise the authors to motivate the use of permutation t-test instead of different statistical tests.

The distributions of these data do not meet the assumptions for use of a parametric t-test. Therefore, our options would be limited to a non-parametric stat such as the Mann Whitney U, Wilcoxon Rank Sum test, or permutation/bootstrapping methods. We have chosen the latter because other non-parametric methods rely on rank-order of the observed variables whereas our data represent continuous magnitude changes more appropriate for permutation testing. Furthermore, these analyses represent a replication of the same approach reported for similarly aged-normal hearing infants.

Paragraph 3.6- p < 0.0001, FDR corrected. Please, check the value of p.

Thank you for noticing this! There was, indeed, one extra zero in the p-value. We have now corrected this as “p < 0.001”.

Discussion: Discussion should start with a summary of the experiment and highlights the principal aspects of novelty, after that I agree with the subparagraphs. BTW, I found it very interesting.

We agree that a discussion overview is needed and have added this to the beginning of the discussion section.

Round 2

Reviewer 2 Report

Thank you for your revision and responses.

I still have some concerns regarding your data analysis and interpretation.

1. This is an MMR study to start with, and you did not find significant results in the MMR activity to address your research questions. So you tried some other analysis such as time-frequency decomposition using Morlet transform (which helped pin down the alpha activity) and spatial PCA using four electrodes (Fz, Cz, M1 and M2). Why is the second step needed? Did the time-frequency analysis also fail to show interesting/significant results to address your research question and therefore you did further PCA analysis? These steps of data analysis seems very exploratory to me. 

2. In my previous comment, I was not questioning the appropriateness of the PCA method, but your specific implementation of PCA by selecting 4 electrodes seemed a bit arbitrary and dubious as those electrodes tend to show polarity reversal in auditory ERPs. Any EEG researcher knows that it is the analysis details that matter. The same analysis method for the same data set can sometimes yield completely different conclusions when different parameters or steps are involved. As far as I can tell, your analysis method seems a bit idiosyncratic. It seems none of the cited studies you included in your response used the specific implementation you described using the four electrodes for PCA. Details were missing regarding the algorithm (e.g., Promax or Varimax). Why not single-trail temporal PCA? How many desired components were specified in the PCA? How much variance did the selected component account for? I encourage you to publicly share your analysis (Matlab?) codes using the osf.io or github platform so that others can check and examine your time-frequency analysis and PCA implementation. 

3. Regarding the source of alpha activity, the SOA would result in a slow rhythm of neural entrainment, which could have harmonics in the alpha range (See

https://www.frontiersin.org/articles/10.3389/fpsyg.2012.00216/full

https://www.frontiersin.org/files/Articles/25026/fpsyg-03-00216-HTML/image_m/fpsyg-03-00216-g002.jpg).

As the infants are more in a wakeful/asleep state during the EEG recording,  the oddball paradigm presumably would trigger some changes in alpha activity. 

4. You argued that the control group data from normal-hearing infants is not necessary for the current study. I don't think that is correct. It is important to see the MMR differences between infants who are hard of hearing and infants who have normal hearing and what results you can get from a control group using the specific analysis you implemented in this study. You mentioned in your response that you had data from normal-hearing infants in another study. If the experimental protocol was the same, you could include the results from normal hearing infants as supplementary materials for comparison. 

Author Response

  1. This is an MMR study to start with, and you did not find significant results in the MMR activity to address your research questions. So you tried some other analysis such as time-frequency decomposition using Morlet transform (which helped pin down the alpha activity) and spatial PCA using four electrodes (Fz, Cz, M1 and M2).

Response: This is incorrect. The MMR has a long history of controversial use partly because of the excessive variability encountered when applying standard/traditional ERP averaging methods. Over the past seven years, our laboratories have been systematically and methodically parsing the sources of variability in this population (infants with hearing loss) with the explicit purpose of narrowing a clinically useful biomarker of early discrimination processes. Our choices of electrodes and frequency band are a direct result of those previous studies in normal hearing infants and are based on our a priori observations, which also corroborate the results from other research groups. While we use an MMR paradigm to elicit these responses, we only include the traditional averages as a point of comparison (and because colleagues and past reviewers have asked us to include these). That is, this is not and never was intended as a traditional MMR analysis, because such analyses are often inconclusive. We never expected the traditional MMR averages to show significant result, nor have we made such claims.

Why is the second step needed? Did the time-frequency analysis also fail to show interesting/significant results to address your research question and therefore you did further PCA analysis? These steps of data analysis seems very exploratory to me. 

Response: Unfortunately, recording multichannel EEG from infants is not optimal for a clinical setting. While it would be great to include multiple analyses from multiple channels, the clinical utility of such an approach is not ideal. Therefore, our goal has been to minimize the amount of information needed to arrive at a single response to determine whether an infant can successfully discriminate between two sounds. PCA is an ideal solution for this approach because it reduces the experimental variability to a single, latent waveform, which is much easier to interpret than comparing multiple, covarying waveforms and is therefore more appropriate for a clinical application.

  1. In my previous comment, I was not questioning the appropriateness of the PCA method, but your specific implementation of PCA by selecting 4 electrodes seemed a bit arbitrary and dubious as those electrodes tend to show polarity reversal in auditory ERPs.

Response: We limited the input to these four electrodes specifically because we have shown that other electrode information does not contribute to the responses (see Gilley et al., 2017 and Uhler et al., 2019). The reviewer is correct that these four electrodes show polarity reversal in auditory ERPs, which is precisely why a PCA of these electrodes is useful. It is easy to show that using a traditional reference (e.g., linked mastoids) is achieved by a weighted average of those electrodes, but with all weights being equal (i.e., the absolute values of those weights are equal, but may differ in their sign). PCA achieves the same goal but with the added benefit of adjusting those weights to maximize the response variability and minimize the random variability that typically leads to inconclusive results.

It seems none of the cited studies you included in your response used the specific implementation you described using the four electrodes for PCA. Details were missing regarding the algorithm (e.g., Promax or Varimax).

Response: This is correct; many of those studies used very large electrode arrays or were implemented with different goals in mind (e.g., source localization via PCA of Laplacian transformed data). My intent of providing those resources was to show that these methods are indeed widely used. Further, we did not use a matrix rotation for this analysis because we wanted to maintain the polarities of the vertex and mastoid electrodes to match the more traditional electrode referencing schemes, which is another benefit of limiting the analysis to these four electrodes.

Why not single-trail temporal PCA?

Response: Single-trial temporal PCA is a great approach that we have experimented with; however, there are still many other details for the recording paradigm that must be examined before such an approach is useable. Further, other methods such as Dynamic Mode Decomposition (DMD) are likely more appropriate for temporal analyses, but more studies are needed to determine appropriate parameters for that approach.

How many desired components were specified in the PCA? How much variance did the selected component account for? I encourage you to publicly share your analysis (Matlab?) codes using the osf.io or github platform so that others can check and examine your time-frequency analysis and PCA implementation. 

Response: We chose a single PCA component (the first component explaining the most variance). Those variances are reported in this manuscript (see Figure 2). We have shared the code for time-frequency transforms via the MathWorks Exchange for MATLAB (https://www.mathworks.com/matlabcentral/fileexchange/56138-aortools?s_tid=prof_contriblnk), although newer versions of MATLAB’s wavelet toolbox easily achieve the same results. Furthermore, we have added this link as supplementary material. PCA via eigendecomposition is also a standard operation included with MATLAB. We do not have code in other languages (e.g., Python) to share at this time.

  1. Regarding the source of alpha activity, the SOA would result in a slow rhythm of neural entrainment, which could have harmonics in the alpha range (See

    https://www.frontiersin.org/articles/10.3389/fpsyg.2012.00216/full

    https://www.frontiersin.org/files/Articles/25026/fpsyg-03-00216-HTML/image_m/fpsyg-03-00216-g002.jpg).

    As the infants are more in a wakeful/asleep state during the EEG recording,  the oddball paradigm presumably would trigger some changes in alpha activity. 

Response: The infants in this study were asleep during the recordings and wakeful periods were rejected as artifact; we only use the EEG obtained during sleep. While it is certainly possible that harmonics of the SOA can appear in the alpha band, this would result in at least 13 harmonics in the alpha range with the lowest being the 10th harmonic of the series (see Hartman 1998)*. The likelihood that so many (high) harmonics would interfere with these responses is very low. Further, even if these harmonics are present in the response, this approach still reveals meaningful differences in these responses.

*W. Hartman, Perception of periodic complex tones, Chapter 6, In Signals, sound, and sensation, Ed, R. Beyer, New York City, Springer-Verlag, 1998, pp 117-130

  1. You argued that the control group data from normal-hearing infants is not necessary for the current study. I don't think that is correct. It is important to see the MMR differences between infants who are hard of hearing and infants who have normal hearing and what results you can get from a control group using the specific analysis you implemented in this study. You mentioned in your response that you had data from normal-hearing infants in another study. If the experimental protocol was the same, you could include the results from normal hearing infants as supplementary materials for comparison.

Response: We disagree with the reviewer that a normal hearing cohort is needed for a within-subjects study design. Our research questions were specifically tailored to hypotheses about hearing aid amplification. Further, our preliminary work on solidifying these methods (Gilley et al., 2017; Uhler et al., 2019) was conducted in normal hearing populations. Therefore, our use of this approach is directly implemented from those methods as a point of comparison. However, there is no direct comparison to a normal hearing group that is appropriate for questions regarding hearing aid use with this study design, and we feel that adding such a comparison would detract from the main purpose of this study.

Reviewer 3 Report

2.6. Statistical Analyses and Hypothesis Testing- I suppose that this paragraph can cause some confusion in the readers. I suggest to organize this section in 2 parts preprocessing and Statistical Analyses and Hypothesis Testing with sub-paragraphs. It is a suggestion.

Indeed, given the complexity of these methods we would like to make everything as clear as possible. In this case, we are struggling to understand the reviewer’s suggestion for changes, but would be more than happy to accommodate if we can receive further/more detailed feedback (e.g., how might we order the sections better to separate the sub-paragraphs, or which information is most confusing that we should focus on, etc.)? Thank you for the suggestions.

Reviewer: I agree with you . Indeed, They are very complex, but it was only a suggestion as  wrote. 

Author Response

Thank you for your comments and suggestions. We will continue to work to simplify this approach in future iterations of this work.